# Rhoifolin Suppresses Cell Proliferation and Induces Apoptosis in Hepatocellular Carcinoma Cells In Vitro and In Vivo

**DOI:** 10.3390/ph18010079

**Published:** 2025-01-10

**Authors:** Ruolan Chen, Zufa Sabeel, Lu Ying, Youfeng Liang, Rui Guo, Mingxuan Hao, Xiaoyang Chen, Wenjing Zhang, Jian Dong, Yan Liu, Changyuan Yu, Zhao Yang

**Affiliations:** Innovation Center of Molecular Diagnostics, College of Life Science and Technology, Beijing University of Chemical Technology, Beijing 100029, China; chen13120171322@126.com (R.C.); zufasabeel183@gmail.com (Z.S.); yl_tlm@163.com (L.Y.); liangyoufeng@163.com (Y.L.); guorui60902@163.com (R.G.); mssjclnl@163.com (M.H.); chenxiaoyang0225@163.com (X.C.); zhangwenjing0818@163.com (W.Z.); dongjian0324@163.com (J.D.); 2024400387@buct.edu.cn (Y.L.)

**Keywords:** rhoifolin, hepatocellular carcinoma, apoptosis, cell cycle arrest, BID

## Abstract

**Background:** Hepatocellular carcinoma (HCC) is the most prevalent malignant tumor, ranking fifth in terms of fatality with poor prognosis and a low survival rate. Rhoifolin (ROF), a flavonoid constituent, has previously been shown to suppress the proliferation of breast and pancreatic cancer cells. However, its inhibitory effect on HCC has remained unexplored. **Objectives:** Exploring the potent inhibitory activities and underlying mechanisms of ROF on HCC cells. **Methods:** The suppressive effect of ROF on HCC cells were assessed via CCK8 assay, apoptosis assay, cell cycle analysis and xenograft tumor mouse model. Furthermore, quantitative real-time PCR and western blot were applied to analyze the underlying mechanisms of ROF on HCC cells. **Results:** Firstly, the IC_50_ values of ROF in HepG2 and HuH7 cells were 373.9 and 288.7 µg/mL at 24 h and 208.9 and 218.0 µg/mL at 48 h, respectively. Moreover, the apoptosis rates of HepG2 and HuH7 cells increased from 6.63% and 6.59% to 17.61% and 21.83% at 24 h and increased from 6.63% and 6.59% to 30.04% and 37.90% at 48 h, respectively. Additionally, ROF induced cell cycle arrest at the S phase in HCC cells. Furthermore, ROF suppressed the tumor growth of HCC cells in vivo without obvious toxicity. Mechanically, ROF facilitated apoptosis by upregulating the expression of PIDD1, CASP8, CASP9, BID, BAX, BIM, and BAK1 in HCC cells. **Conclusions:** ROF significantly restrains the growth of HCC cells in vitro and in vivo, which could be an effective supplement for HCC therapy.

## 1. Introduction

Globally, hepatocellular carcinoma (HCC) is the most prevalent malignant tumor, ranking fifth in terms of fatality [1]. The main treatments for HCC are surgery, radiotherapy, drug therapy, and immunotherapy [2]. The aforementioned treatments are not applicable to all cases of HCC with a poor prognosis [3], low survival rate [4], and recurrence tendency.

In recent years, there has been a growing interest in the potential of Traditional Chinese medicine (TCM) to inhibit cancer growth [5,6]. A substantial body of scientific evidence attests to the antimicrobial, anti-inflammatory, antioxidant, and anticancer properties of TCM [7]. TCMs have been shown to exhibit anticancer effects, targeting multiple pathways and causing few side effects. TCMs effectively prevent primary HCC by enhancing cellular immune function and reducing the occurrence of tumor immune escape [8]; inhibiting chronic inflammatory processes and the release of pro-inflammatory cytokines; controlling signaling pathways and influencing the expression of cell cycle proteins; and other mechanisms. For example, Glabridin has been demonstrated to inhibit the growth of urothelial bladder carcinoma cells in both in vitro and in vivo models. This is achieved through the induction of cell apoptosis and cell cycle arrest [9]. The anticancer attributes of cantharidin are involved in a number of molecular mechanisms and pathways [10]. Betulin has been demonstrated to exert a beneficial effect in the context of 7,12-dimethylbenz(a)anthracene-induced rat mammary cancer, with its action involving the modulation of the mitogen-activated protein kinase (MAPK) and aryl hydrocarbon receptor/nuclear factor erythroid 2-related factor (AhR/Nrf-2) signaling pathways [11]. The flavonoid medicarpin has been demonstrated to induce G1 arrest and the mitochondria-mediated intrinsic apoptotic pathway in bladder cancer cells [12]. Rodopsin has been demonstrated to possess anti-inflammatory, antioxidant, hepatoprotective, and anticancer effects and has been shown to prevent liver damage caused by a variety of factors [13].

It has been demonstrated that flavonoids possess a range of anticancer properties [14]. Flavonoids have the capacity to stimulate apoptotic pathways and to downregulate pro-inflammatory signaling pathways [15]. For example, isocoumarin, a flavonoid derived from carbohydrates, can be synthesized from 3-glycosyl isocoumarin and 3-glucosyl-substituted isocoumarin [16]. Isocoumarins have been demonstrated to possess potent antioxidant and anti-inflammatory activities, which are essential for the treatment of chronic diseases [17].

ROF is a flavonoid component [18]; the molecular structure of ROF is shown in Figure 1. ROF treatment at concentrations of 10 μM and 40 μM was observed to significantly inhibit the migration of breast cancer cells [19]. ROF has been demonstrated to mitigate ETH-induced liver injury by impeding NF-κB phosphorylation [20]. Furthermore, ROF has been shown to inhibit the proliferation and promote the apoptosis of pancreatic cancer cells, as well as inhibit cell migration and invasion and enhance the antioxidant capacity of PANC-1 and ASPC-1 [21].

The objective of this study was to assess the inhibitory effect of ROF on HCC and to elucidate the underlying mechanism promoting apoptosis. This was achieved through growth inhibition assay, cell cycle analysis, apoptosis assay, quantitative real-time PCR, and Western blotting analysis in order to observe the underlying molecular mechanisms. The effect of ROF was further validated in subcutaneous tumor transplantation experiments in HCC mice, and its safety was confirmed through histochemical experiments (HEs).

## 2. Results

### 2.1. ROF Inhibited the Proliferation of HepG2 and HuH7

To evaluate the inhibitory effect of ROF on HCC cells, a cell proliferation assay was performed. ROF exhibited a time-dependent inhibitory effect on cell viability. The IC_50_ value of ROF in HepG2 cells was 373.9 µg/mL at 24 h and 208.9 µg/mL at 48 h (Figure 2a). For HuH7 cells, the IC50 value was 288.7 µg/mL at 24 h and 218.0 µg/mL at 48 h (Figure 2b). 

### 2.2. ROF-Induced Apoptosis of HepG2 and HuH7

The HepG2 cells revealed a notable increase in apoptotic cells following a 48 h treatment period. In HepG2 cells, the apoptotic rate increased from 6.63% to 17.61% at 100 µg/mL and further to 30.04% at 200 µg/mL (Figure 3a,b). Similarly, in HuH7 cells, the apoptosis rate rose from 6.59% to 21.83% at 100 µg/mL and to 37.90% at 200 µg/mL (Figure 3a,c). These findings suggest a dose-dependent increase in apoptosis following ROF treatment in both HepG2 and HuH7 cells.

### 2.3. ROF-Induced Cell Cycle Arrest in HepG2 and HuH7 HCC Cells

The impact of ROF on cell cycle progression in HCC cells was assessed using a cell cycle assay, which demonstrated significant S-phase arrest. In HepG2 cells treated with ROF, the S-phase proportion increased from 39.71% to 47.60% at 100 µg/mL and from 39.71% to 51.84% at 200 µg/mL (Figure 4a,b). In HuH7 cells, the S phase increased from 30.31% to 37.90% and from 30.31% to 41.51% at 200 µg/mL (Figure 4c,d). The aforementioned results demonstrate that ROF effectively induces S-phase arrest, thereby inhibiting cellular proliferation in HCC cells.

### 2.4. ROF Inhibits Tumor Growth In Vivo

To further investigate the molecular mechanism of ROF through which it promotes HCC cell apoptosis, we conducted qRT-PCR and WB experiments. The ROF treatment resulted in a significant increase in the expression of PIDD1 in both HepG2 and HuH7 cells, with a 6.80-fold and 2.49-fold increase, respectively. Additionally, ROF significantly elevated the expression of several Bcl-2 family members, including BID, BAX, BAK1, and BIM (Figure 5a,b). Among these, BID showed the most notable increase (14.58-fold), while other genes displayed increases ranging from 1.35- to 2.64-fold.

Further, caspase family proteins (CASP8/9) manifested considerable expression in response to ROF treatment, with a 1.75- and 1.66-fold increase observed in HepG2 cells and a 2.65- and 3.15-fold increase in HuH7 cells, respectively. The protein expressions of BID, BIM, BAX, CASP8, and BAK1, which promote apoptosis, were all increased in response to ROF treatment (Figure 6a,b).

To further verify the inhibitory effect of ROF on HCC, a subcutaneous tumor transplantation assay was performed in HCC mice (Figure 7a). The grouping was conducted when the tumor volume reached 97.59 mm^3^. By day 21, the average tumor volume in the ROF-treated group (HuH7 cells) was 390.75 mm^3^, compared to 1062.53 mm^3^ in the control group (Figure 7b–d). Additionally, the experimental group’s average tumor volume was 250.78 mm^3^, while the control group’s tumor volume increased to 833.71 mm^3^, indicating that ROF significantly suppressed tumor growth in vivo. The mean tumor volume of the mice provides evidence that ROF can markedly suppress tumor growth in vivo. The results of histopathological evaluation showed that there were no lesions or abnormalities in the organs of the ROF-treated or control mice (Figure 7e). These results confirm that ROF was non-toxic to mice and confirm the safety of ROF.

## 3. Discussion

HCC represents a serious public health concern, affecting the global population. With advances in medical and biological research, the exploration and validation of natural products with significant biological activities has become increasingly prevalent. This study is the first to investigate the inhibitory effect of ROF on HCC cells and its mechanism of apoptosis promotion. The experimental results demonstrated that ROF exhibited potent inhibitory activity against HCC cells.

As illustrated in Table 1, ROF exhibited a lower IC_50_ value, indicating high potency and allowing effective inhibition at lower concentrations than other compounds. This characteristic is of particular significance in the development of pharmaceuticals, as it suggests that ROF could achieve therapeutic effects at lower doses, potentially reducing the incidence of adverse effects. Furthermore, the lower IC_50_ value of ROF may also indicate a higher affinity for its target, which could be a crucial factor in its mechanism of action and therapeutic potential.

ROF was found to induce apoptosis in HCC cells, with a significant increase in apoptotic cells observed in both HepG2 and HuH7 cells in a dose-dependent manner. Moreover, the inhibitory effect of ROF on HCC cells is reflected in the cell cycle, whereby the cell cycle is arrested in the S phase. This suggests that ROF may interact with the signaling pathways that regulate the progression of the cell cycle, thereby interfering with the cell cycle and preventing HCC cells from passing through the S phase. This ultimately results in the inhibition of HCC cell growth and proliferation. The period of cell cycle inhibition is variable. Other compounds, such as Tacoside, exhibit similar effects, causing cell cycle arrest in the G2/M phase and inducing caspase-dependent apoptosis in HCC cells [25].

The results of quantitative real-time PCR show a significant increase in the expression of pro-apoptotic proteins, including PIDD1, BID, BAX, BAK1, and BIM, as well as caspase family proteins (CASP8/9). Caspase family proteins (CASP8/9) are cysteine proteases that act as key enzymes in the apoptosis signaling pathway. Their elevated expression can lead to the activation of the apoptotic pathway, resulting in an overall increase in apoptosis. ROF induces apoptosis in HCC cells via an intramitochondrial pathway (Figure 8).

These proteins can promote apoptosis by forming pores in the mitochondrial membrane, leading to an increase in mitochondrial membrane permeability and the release of apoptotic factors such as cytochrome C. Previous studies, such as one on the flavonoid medicarpin, show similar results, where apoptosis in glioblastoma cells was triggered via the upregulation of BID, BAX, CASP3, CASP8, and CYCS [26]. Of course, there are other apoptotic pathways, such as the PI3K/AKT pathway downregulated by dehydrocostus lactone (DHL) in HepG2 cells; the mitochondrial apoptosis pathway offers the advantage of multilevel regulation, associating energy metabolism with apoptosis and responding to diverse signals.

Subcutaneous transplantation in mice with HCC showed that ROF markedly suppressed tumor growth by inducing apoptosis and inhibiting cell cycle progression. Histopathological examination revealed no adverse effects on normal organs, suggesting that ROF is safe for healthy tissues. The minimal risk of significant side effects in patients supports its safety and tolerability. This selective toxicity indicates ROF’s potential as a targeted therapeutic agent. These findings provide substantial support for the further research and development of ROF-based therapeutic strategies and increase its potential for clinical application in HCC treatment.

Overall, these results indicate that ROF suppresses cell proliferation and induces apoptosis in HCC cells both in vitro and in vivo. Further research on the molecular mechanisms by which ROF inhibits HCC cells may facilitate the development of more effective combination therapies for HCC. Nevertheless, further research is needed to determine the precise target of ROF and the mechanism by which it exerts its function. Exploring the specific pathways influenced by ROF may also provide valuable insights into innovative therapeutic strategies for other cancers with similar cellular and molecular attributes.

## 4. Materials and Methods

### 4.1. Cell Lines and Reagents

ROF (HPLC >98.0%, Desite Biotechnology, Chengdu, China), dissolved in dimethyl sulfoxide (DMSO) (Solarbio, Beijing, China), and stored at −20 °C. The HCC cell lines (HepG2 and HuH7) and normal liver epithelial cell lines (L02) were provided by Procell Life Science & Technology (Wuhan, China), and were cultured in DMEM (HyClone, Logan, UT, USA) medium with 10% FBS (Gibco, Waltham, MA, USA) and 1% PS (Gibco, Waltham, MA, USA) at 37 °C in 5% CO_2_. Once the cells had reached 90% confluency, 1 mL of trypsin (Biosharp, Hefei, China) was added, and cells were allowed to soak for a few minutes to separate the adherent cells from the T25 flasks. The medium was changed to a 1:3 ratio two or three times per week.

### 4.2. Cell Proliferation Assay

The experiment was divided into a control group (DMSO) and a treatment group (ROF). HepG2, HuH7, and L02 cells were seeded into 96-well culture plates at a seeding density of 5 × 10^3^ cells per well. The cells in the treatment group were incubated with ROF at concentrations of 50, 150, 250, 300, 350, 450, and 600 µg/mL for either 24 h or 48 h. After incubation, the culture medium was replaced with fresh medium containing 10% CCK8 reagent (Beyotime Biotechnology, Shanghai, China) and incubated for 2 h at 37 °C. Absorbance was measured at 450 nm using a Multiskan FC microplate reader (51119080, Thermo Fisher Scientific, Waltham, MA, USA). Data were analyzed with GraphPad Prism software 8.0 to calculate IC_50_ values and cell viability. In the proliferation experiment, the cell numbers of the DMSO group were identified as 100%. On this basis, the cell numbers of the ROF group were converted into a relative percentage compared with those of the DMSO group. Cell viability was calculated using the following formula:Cell viability (%) = [(A experimental well − A blank well)/(A control well − A blank well)] × 100%

### 4.3. Cell Cycle Analysis

HepG2 and HuH7 cells were seeded at a seeding density of 2 × 10^5^ cells per well in a 6-well culture plate for cell cycle analysis. After the cells adhered to the wells, they were cultured in 1640 and DMEM media without FBS for 24 h to ensure a consistent cell cycle. The cells were then treated with 100 or 200 µg/mL ROF for 48 h, after which they were harvested and stored overnight at −20 °C in 500 µL 70% ethanol. The cells were washed 3 times with PBS and resuspend in 500 µL PBS. In total, 50 µg/mL of RNase A and 25 µL PI (1 mg/mL) were added to the cell suspension (Beyotime Biotechnology, Shanghai, China). The cell cycle distribution was then analyzed using BD Biosciences (San Jose, CA, USA) at an excitation wavelength of 561 nm. FlowJo software 10.4 was used to calculate the percentage of cells in each phase of the cell cycle.

### 4.4. Apoptosis Assay

HepG2 and HuH7 cells were seeded at a density of 2 × 10^5^ cells per well in a 6-well culture plate for apoptosis detection. Once the cells adhered to the bottom of the well, they were treated with 100 or 200 µg/mL ROF for 48 h. After the treatment period, the cells were collected and stained with 5 µL Annexin V-FITC and 10 µL PI solution (Beyotime Biotechnology, Shanghai, China) and incubated in the dark at room temperature for 30 min. After staining, apoptosis was assessed using the Calibur flow cytometer (BD Biosciences, Franklin Lakes, NJ, USA) with detection at 488 nm and 535 nm. FlowJo_V10 software was used to calculate the percentage of cells in different apoptotic phases.

### 4.5. Quantitative Real-Time PCR

HepG2 and HuH7 cells were seeded at a density of 2 × 10^5^ cells per well in a 6-well culture plate for apoptosis detection. After the cells adhered to the wall of the well, HepG2 and HuH7 cells were treated with 200 µg/mL ROF for 48 h. RNA was extracted from the cells using an RNA Kit and reverse-transcribed into cDNA using a Reverse Transcription Kit. The RNAprep pure cell kit and FastKing RT kit were purchased from Tiangen, China. Quantitative real-time PCR was performed using the SuperReal PreMix Plus (SYBR Green, TIANGEN Biotech, Beijing, China) Kit (FP205, TIANGEN Biotech, Beijing, China) to measure the relative expression of genes. Finally, the QuantStudio1 real-time PCR system (A40425, Thermo Fisher Scientific, Waltham, MA, USA) was used. The relative mRNA levels of each gene were calculated using the 2^−ΔΔCt^ method, and the primer sequences are listed in Table 2.

### 4.6. Western Blot Analysis

HepG2 and HuH7 cells were seeded at a density of 2 × 10^5^ cells per well in a 6-well culture plate for apoptosis detection. Once the cells adhered to the walls of the wells, they were treated with 200 µg/mL ROF for 48 h. Following the treatment, the cells were processed for protein electrophoresis (SDS-PAGE). Immediately after transfer to the membrane, the protein bands were transferred to a PVDF membrane (Millipore, Bedford, MA, USA). After membrane transfer, 7% nonfat dry milk was used for protein blocking, and after blocking, 7% nonfat dry milk containing antibodies was used for incubation overnight at 4 °C. After completing the incubation, the PVDF membrane was washed three times with 1× TBST. It was then incubated with 7% nonfat dry milk containing the secondary antibody. Following this, the membrane was washed three times with 1× TBST. The target protein was visualized using an enhanced chemiluminescence (ECL) kit (PE0010, Solarbio, Beijing, China) and then analyzed using ImageJ 2.1.0. β-Actin was used as an internal control.

### 4.7. Experiments on Subcutaneous Transplantation of HCC Cells

BALB/c nude mice (20 females, 6 weeks old, 15 ± 2 g, SiPeifi, Beijing, China) were maintained under specific-pathogen-free identical conditions. HepG2 and HuH7 cells (5 × 10^6^) were injected subcutaneously into the right shoulder of the mice. When the tumors grew to a size of 100 mm^3^, the mice were randomly divided into two groups: the experimental group (ROF, 10 mg/kg [27], 100 μL, *n* = 5) and the control group (1% DMSO, 100 μL, *n* = 5). The treatments were administered intraperitoneally every 3 days. ROF was dissolved in DMSO (50 mg/mL) and stored at −20 °C.

### 4.8. Histopathological Analysis

Pathological sections were prepared from the major organs of experimental and control mice. Paraffin sections were first dewaxed by placing them in an environmentally safe dewaxing solution, followed by soaking in anhydrous ethanol and finally rinsing with distilled water. The sections were then stained as follows: hematoxylin for approximately 3 min, rinsed with tap water; hematoxylin differentiation solution for a few seconds, rinsed with tap water; and hematoxylin blue solution, followed by rinsing with running water. For final dehydration and sealing, the sections were sequentially placed in 75% alcohol for 5 min, 85% alcohol for 5 min, and anhydrous ethanol for 5 min. The sections were then removed from the xylene, gently dried to complete dehydration, and sealed with glue. The stained sections were observed microscopically under white light to analyze the results.

## 5. Conclusions

This study provides compelling evidence that ROF significantly inhibits HCC cell growth by inducing apoptosis and arresting the cell cycle in the S phase. ROF’s low IC_50_ value underscores its high potency, suggesting potential therapeutic advantages at lower doses with minimized side effects. The in vivo results in HCC mice demonstrated ROF’s ability to selectively target tumor cells while sparing normal tissue, highlighting its safety and targeted efficacy. These findings support ROF’s potential as a promising candidate for HCC therapy, warranting further investigation to refine its mechanism and explore its use in combination therapies for HCC and potentially other cancers.

## Figures and Tables

**Figure 1 pharmaceuticals-18-00079-f001:**
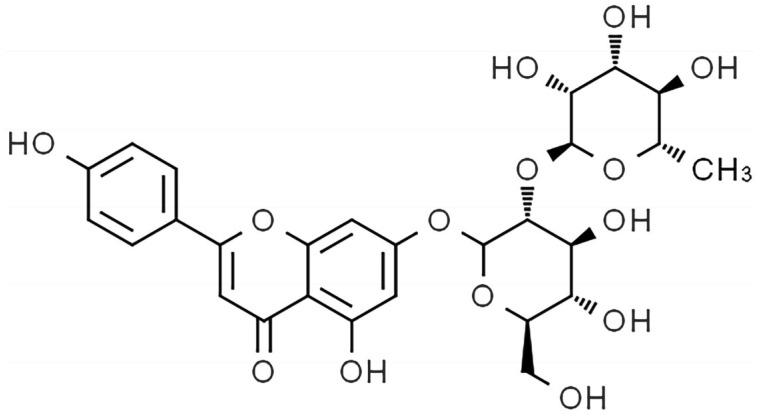
Chemical structure of ROF.

**Figure 2 pharmaceuticals-18-00079-f002:**
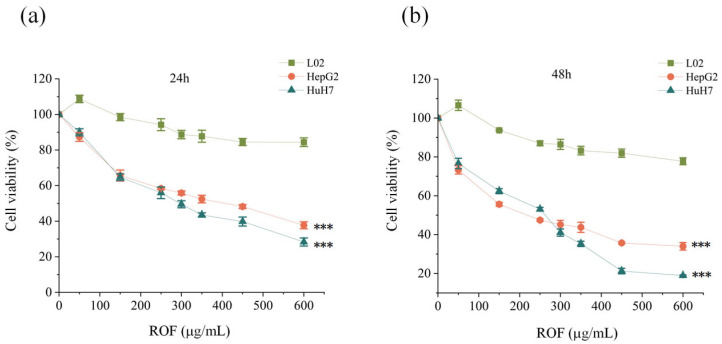
Effect of ROF on the inhibition of HCC cell proliferation evaluated using the CCK8 assay. (**a**) HepG2 and HuH7 cells were treated with ROF for 24 h. Cell viability was detected by CCK-8 assay. (**b**) HepG2 and HuH7 cells were treated with ROF for 48 h. Cell viability was detected by CCK-8 assay. (*n* = 3, *** *p* < 0.001).

**Figure 3 pharmaceuticals-18-00079-f003:**
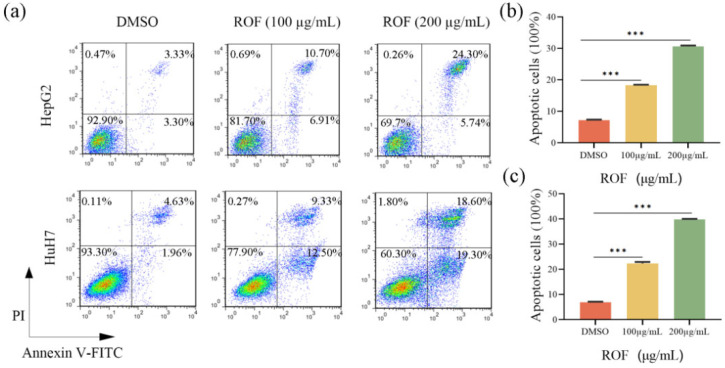
Apoptosis-inducing effect of ROF on HepG2 and HuH7 cells. (**a**) Annexin-V-FITC/PI double-staining assay was used to analyze cell apoptosis by flow cytometry. (**b**) Histogram of apoptosis statistics of HepG2 cells. (**c**) Histogram of apoptosis statistics of HuH7 cells. (*n* = 3, *** *p* < 0.001).

**Figure 4 pharmaceuticals-18-00079-f004:**
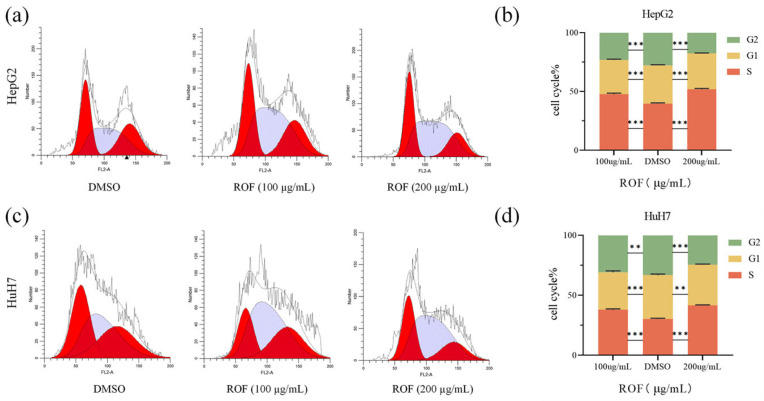
ROF can affect HepG2 and HuH7 cell cycles. (**a**) Flow cytometry detected the cell cycle distribution of HepG2 cells. (**b**) Histogram of cell cycle of HepG2 cells. (**c**) Flow cytometry detected the cell cycle distribution of HuH7 cells. (**d**) Histogram of cell cycle of HuH7 cells. (*n* = 3, ** *p* < 0.01, *** *p* < 0.001).

**Figure 5 pharmaceuticals-18-00079-f005:**
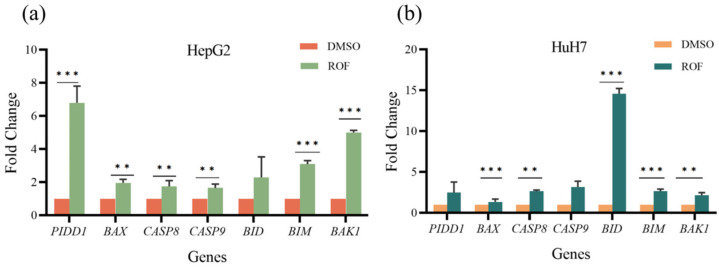
qRT-PCR experimental results after ROF treatment of HepG2 and HuH7 cells. (**a**) Histogram of gene expression of HepG2 cells. (**b**) Histogram of gene expression of HuH7 cells. (*n* = 3, ** *p* < 0.01, *** *p* < 0.001).

**Figure 6 pharmaceuticals-18-00079-f006:**
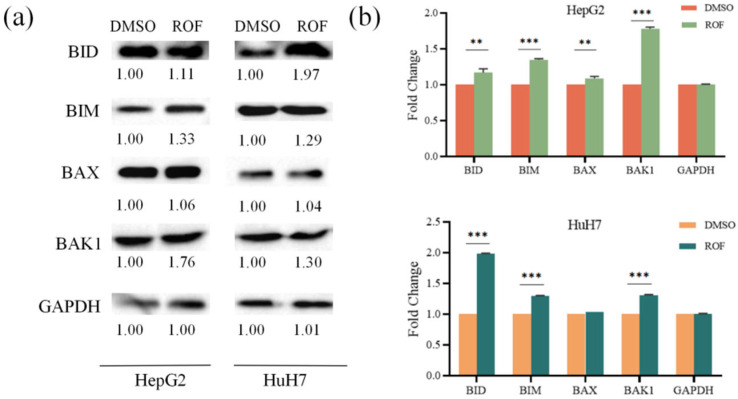
Western blotting analysis of the protein expressions after ROF treatment of HepG2 and HuH7 cells. (**a**) Histogram of protein expression of HepG2 cells. (**b**) Histogram of protein expression of HuH7 cells. (*n* = 3, ** *p* < 0.01, *** *p* < 0.001).

**Figure 7 pharmaceuticals-18-00079-f007:**
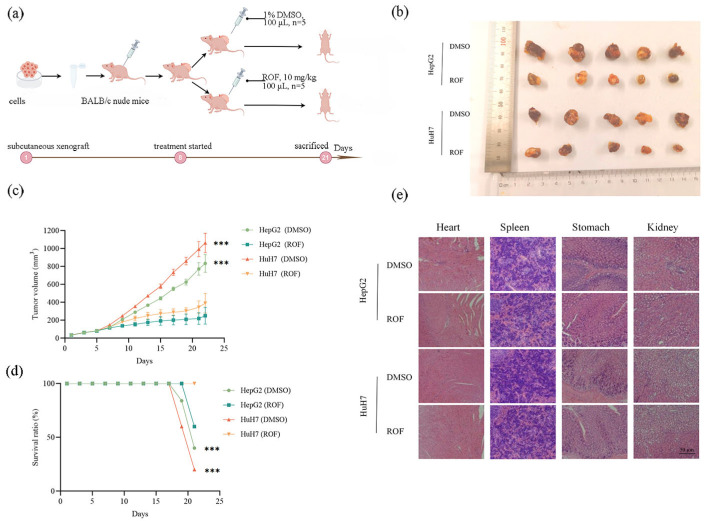
In vivo anti-tumor effects of ROF on nude mice xenografts of HepG2 and HuH7 cells. (**a**) The schedule of the whole animal experiment. (**b**) Schematic diagram of excised tumors. (**c**) Schematic diagram of tumor volume in mice. (**d**) Schematic diagram of the number of surviving mice. (**e**) Histopathological analysis of the major organs of the heart, spleen, lungs, and kidneys of mice. (*n* = 3, *** *p* < 0.001).

**Figure 8 pharmaceuticals-18-00079-f008:**
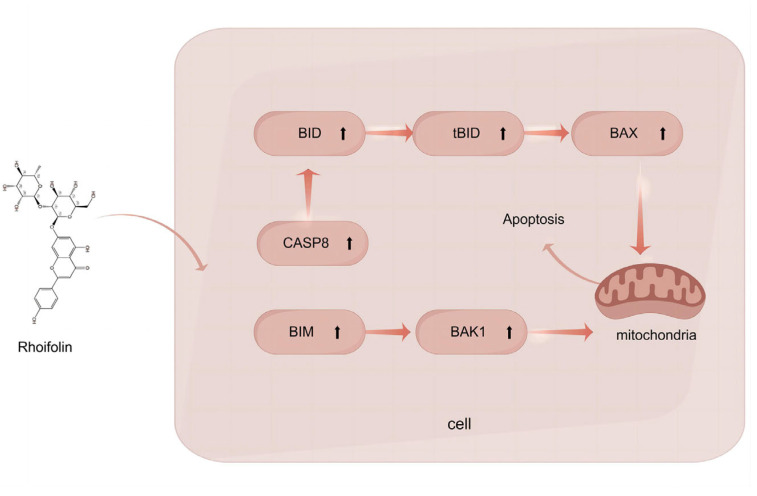
The molecular mechanisms of ROF on HCC cells. ROF facilitated apoptosis by upregulating the expression of CASP8, BID, BAX, BIM and BAK1 in HCC cells.

**Table 1 pharmaceuticals-18-00079-t001:** Treatment of HCC with Chinese medicine monomers.

Name	Cell Line	IC_50_ (24 h)	IC_50_ (48 h)	Mechanism of Action	References
Baicalin	HepG2	89.27 µg/mL	89.27 µg/mL	Baicalin elevated ROS levels in HepG2 cells, downregulated p-PI3K/PI3K, p-Akt/Akt, and p-FoxO3a/FoxO3a proteins, and promoted iron death in HepG2 cells.	[22]
Solanum melongena glycoalkaloids	HuH7	9.81 µg/mL 11.14 µg/mL	NA	The isolated glycoalkaloids induced an antiproliferative effect which is attributed to the inhibition of cell cycle progression in the S phase.	[23]
Ixeris sonchifolia extract	HepG2	2.5 µg/mL	0.5 µg/mL	Ixeris sonchifolia extract induces hepatocellular carcinoma apoptosis via the PI3K/AKT pathway.	[24]
Rhoifolin	HepG2 and HuH7	373.9 µg/mL and 288.7 µg/mL	208.9 µg/mL and 218.0 µg/mL	BID, BIM, BAX, CASP8, and BAK1, which promote apoptosis, were all increased.	This study

**Table 2 pharmaceuticals-18-00079-t002:** Primer list of genes used for qRT-PCR.

Gene	Forward Primer Sequence (5′-3′)	Reverse Primer Sequence (5′-3′)
GAPDH	AAGGTGAAGGTCGGAGTCAA	GGAAGATGGTGATGGGATTT
PIDD1	GAGCCTCGTCGAGTCTCCAT	GGCCCAGTACAACAGGTGC
BAX	CCTTTTGCTTCAGGGTTTCA	CAGTTGAAGTTGCCGTCAGA
CASP8	TTTGACCACGACCTTTGAAGAG	CCCCTGACAAGCCTGAATAAAAA
CASP9	CGAACTAACAGGCAAGCA	AATCCTCCAGAACCAATGTC
BID	ATGGACCGTAGCATCCCTCC	GTAGGTGCGTAGGTTCTGGT
BIM	CTGAGTGTGACCGAGAAG	GATTACCTTGTGGCTCTGT
BAK1	TCTGGCCCTACACGTCTACC	ACAAACTGGCCCAACAGAAC

## Data Availability

The data that support the findings of this study are available from the corresponding author by E-mail upon reasonable request.

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
