# Peer review of "Rhoifolin Suppresses Cell Proliferation and Induces Apoptosis in Hepatocellular Carcinoma Cells In Vitro and In Vivo"

_pharmaceuticals, 2025, doi:10.3390/ph18010079_

Round 1

Reviewer 1 Report

Comments and Suggestions for Authors

The article titled “Rhoifolin suppresses cell proliferation and induces apoptosis in hepatocellular carcinoma cells in vitro and in vivo” presents a study on the effects of Rhoifolin (ROF), a flavonoid compound, on hepatocellular carcinoma (HCC) cells. The research demonstrates that ROF inhibits HCC cell proliferation, induces apoptosis, and arrests the cell cycle in the S phase. The study employs various experimental techniques, including cell proliferation assays, apoptosis assays, cell cycle analysis, qRT-PCR, and Western blotting. ROF was found to upregulate pro-apoptotic proteins such as PIDD1, BID, BAX, BAK1, BIM, and caspase family proteins (CASP8/9). In vivo experiments using HCC mouse models confirmed ROF's ability to suppress tumor growth without causing toxicity to normal tissues. The moderate IC50 values of ROF suggest its potency and potential for therapeutic use at lower doses. Overall, this research provides preliminary evidence for ROF's potential as a promising candidate for HCC therapy, warranting further investigation into its precise molecular targets and mechanisms of action. However, there are some limitations and areas for improvement in the study. The reviewer has the following comments and suggestions to improve the quality and clarity of paper

1.     The reported IC50 values of ROF, ranging from 208.9 to 373.9 µg/mL, appear to be higher compared to those of known chemotherapeutics or the monomers listed in Table 2. It is recommended that the authors present all IC50 values of the compounds compared in consistent units, making it easier for readers to interpret and compare the data effectively.

2.     The authors should clarify the rationale for selecting a 10 mg/kg dose for the in vivo studies. Considering that ROF is a small molecule, did the authors perform pharmacokinetic studies to determine how long the compound remains in circulation? Providing this information would enhance the understanding of the dosing strategy.

3.     The study primarily focuses on ROF, without delving into its broader applicability to other flavonoids or plant secondary metabolites. Expanding the introduction to include these aspects could greatly enhance the study's relevance and scientific impact. For instance, isomers of flavonoid phenolic compounds, such as isocoumarins, are well-known for their diverse biological activities, including antioxidant, antimicrobial, and anticancer properties. The following references offer valuable insights into these compounds and their potential applications. Incorporating these studies into the introduction would provide a broader context, highlighting the therapeutic relevance of ROF and related derivatives across various applications.

https://www.sciencedirect.com/science/article/pii/S0223523416307243

https://chemistry-europe.onlinelibrary.wiley.com/doi/abs/10.1002/ejoc.201601264

4.     The quality of the figures in the article does not meet the journal's standards. The authors are advised to provide higher-resolution versions of the figures to ensure clarity and adherence to publication requirements. The manuscript contains several typographical and spelling errors that need to be addressed. The authors are encouraged to thoroughly review the entire manuscript to correct these mistakes. For example, in Line 59, and in Line 237.

5.     The authors are encouraged to review the formatting of the cited references to ensure consistency. All references, including journal titles and author names, should adhere uniformly to the journal's prescribed style guidelines.

Author Response

Dear Reviewer,

Attached please find the revised version of our manuscript entitled “Rhoifolin suppresses cell proliferation and induces apoptosis in hepatocellular carcinoma cells in vitro and in vivo” (pharmaceuticals-3379644). We appreciated very much for valuable comments and helpful suggestions from you and the reviewers, which have guided us to significantly improve the quality of our manuscript. We have thoroughly revised the manuscript accordingly, and the changes were highlighted in blue in the revised version.

Question 1. The reported IC50 values of ROF, ranging from 208.9 to 373.9 µg/mL, appear to be higher compared to those of known chemotherapeutics or the monomers listed in Table 2. It is recommended that the authors present all IC50 values of the compounds compared in consistent units, making it easier for readers to interpret and compare the data effectively.

Reply 1:

We have made the revisions as suggested and all IC50 values in Table 2 are presented in micrograms per milliliter (µg/mL), allowing for a direct comparison of the inhibitory effect of ROF with known chemotherapeutic agents and its monomers. We believe this adjustment will greatly assist readers in understanding the comparative effectiveness of ROF with other compounds.Thank you again for your suggestion, which has helped us to make our research results clearer and more comparable.

Revised Table 2 . Treatment of HCC with Chinese medicine monomers.(pages 9, line 260)

Name

cell line

IC50 (24h)

IC50 (48h)

mechanism of action  

References

Baicalin

HepG2

89.27 µg/mL

89.27 µg/mL

Baicalin elevated ROS levels in HepG2 cells, down-regulated p-PI3K/PI3K, p-Akt/Akt, and p-FoxO3a/FoxO3a proteins, and promoted iron death in HepG2 cells.

[20]

Solanum melongena glycoalkaloides

HuH7

9.81 µg/mL

11.14 µg/mL

NA

Te isolated glycoalkaloids induced antiproliferative elect which is attributed to inhibiting cell cycle progression in S-phase;

[21]

Ixeris sonchifolia 

extract

HepG2

2.5 µg/mL

0.5 µg/mL

Ixeris sonchifolia extract Induces Hepatocellular Carcinoma Apoptosis via the PI3K/AKT Pathway

[22]

Rhoifolin

HepG2 and HuH7

373.9 µg/mL and  

288.7 µg/mL

208.9 µg/mL

and

218.0 µg/mL

BID, BIM, BAX, CASP8, and BAK1, which promote apoptosis, were all increased

This study

Question 2. The authors should clarify the rationale for selecting a 10 mg/kg dose for the in vivo studies. Considering that ROF is a small molecule, did the authors perform pharmacokinetic studies to determine how long the compound remains in circulation? Providing this information would enhance the understanding of the dosing strategy.

Reply 2:Due to the constraints of experimental time and conditions, this article did not conduct pharmacokinetic studies. The 10 mg/kg dose was selected for in vivo studies on the basis of the results of existing literature and dose conversion from animal models.

Anroop Nair, et. al. reported that Rhoifolin exerts a protective effect against alcoholic liver disease in vivo and in vitro by inhibiting the TLR4/NF-κB signalling pathway (Front Pharmacol. 2022.13:878898). To investigate the efficacy of Rhoifolin, three ROF treatment groups (10, 20 and 40 mg/kg) were applied in specific experiments. The findings of the study demonstrated that a low concentration of ROF (10 mg/kg) was efficacious in mitigating ETH-induced liver injury in mice. Conversely, high concentrations of ROF (20 and 40 mg/kg) resulted in cell or organ damage. Therefore, it is recommended that ROF not be administered at a dosage exceeding 20 mg/kg.

Furthermore, Chen found that 231 mg/kg ROF treatment for 24 hours significantly affected the cell viability of rat chondrocytes (Osteoarthritis Cartilage. 2022.30(5):735-745). However, 11.5 mg/kg ROF had no cytotoxic effect on chondrocytes within 24-48 hours.These findings indicated that ROF had no discernible side effects on the organs when administered at doses below 11.5 mg/kg.

Additionally, our group found that 10 mg/kg glabridin could suppress the growth of UBC cells in vivo by triggering apoptosis and cell cycle arrest in mice (Chem Biol Drug Des. 2023. 101(3):581-592), while exhibiting no discernible impact on normal tissues and organs.

In order to minimise side effects of ROF on normal tissues in vivo, 10 mg/kg ROF was selected in mice model experiments in this study. The results indicated that 10 mg/kg ROF could effectively suppress the tumor growth of HCC cells and there were no significant side effects in vivo (Figure 7).

Question 3. The study primarily focuses on ROF, without delving into its broader applicability to other flavonoids or plant secondary metabolites. Expanding the introduction to include these aspects could greatly enhance the study's relevance and scientific impact. For instance, isomers of flavonoid phenolic compounds, such as isocoumarins, are well-known for their diverse biological activities, including antioxidant, antimicrobial, and anticancer properties. The following references offer valuable insights into these compounds and their potential applications. Incorporating these studies into the introduction would provide a broader context, highlighting the therapeutic relevance of ROF and related derivatives across various applications.

https://www.sciencedirect.com/science/article/pii/S0223523416307243

https://chemistry-europe.onlinelibrary.wiley.com/doi/abs/10.1002/ejoc.201601264

Reply 3:

Thank you for your valuable suggestions regarding our study. We acknowledge the importance of expanding the focus of our research on ROF to include the broader applicability to other flavonoids and plant secondary metabolites, which would greatly enhance the relevance and scientific impact of our study. The below information have been supplemented as follows:

For example, isocoumarin, a flavonoid derived from carbohydrates, can be synthesised from 3-glycosyl isocoumarin and 3-glucosyl-substituted isocoumarin[16]. Isocoumarins have been demonstrated to possess potent antioxidant and anti-inflammatory activities, which are essential for the treatment of chronic diseases[17]. (page 2, line 56-60)

Reference in the manuscript

[16]Sudarshan, K. and I.S. Aidhen, Convenient Synthesis of 3-Glycosylated Isocoumarins.2017. 2017(1): p. 34-38.

[17]Ramanan, M., et al., Inhibition of the enzymes in the leukotriene and prostaglandin pathways in inflammation by 3-aryl isocoumarins. Eur J Med Chem, 2016.124: p. 428-434.

Question 4. The quality of the figures in the article does not meet the journal's standards. The authors are advised to provide higher-resolution versions of the figures to ensure clarity and adherence to publication requirements.

The manuscript contains several typographical and spelling errors that need to be addressed. The authors are encouraged to thoroughly review the entire manuscript to correct these mistakes. For example, in Line 59, and in Line 237.

Reply 4:

Thank you for your valuable comments and guidance. We have taken note of the issues regarding the quality of figures, typesetting, and spelling in our article, and we sincerely apologize for these shortcomings. We will immediately implement the following measures to improve our manuscript:

Figure Quality Improvement: We have resubmitted all figures, ensuring they are submitted at a higher resolution to meet the journal's standards.

Typesetting and Spelling Error Correction: We have conducted a comprehensive proofreading of the entire manuscript to correct typesetting and spelling errors, ensuring the accuracy and professionalism of the text. For example,

The phrase "with a molecula formula [14] as illustrated in Fig.1" has been revised to " is shown in Figure 1". (page 2, line 61)

The word “promotion” has been revised to “promoting”. (page2 , line 72)

The word “through ” has been revised to “validated in”. (page2 , line 75)

The word “have” has been revised to “had”. (page2 , line 85)

The word “using” has been revised to “with”. (page3 , line 97)

The phrase “wall behind the " has been revised to “were seeded at”. (page3 , line 104)

The word “and the " has been revised to “after which”. (page3 , line107)

The word “ wash” has been revised to “washed”. (page3 , line109)

The word “detected” has been revised to “analyzed”. (page3 , line111)

The word “staining” has been revised to “stained”. (page3 , line118)

The phrase “was used to detect” has been revised to “ detection”. (page3 , line121)

The word “rinsed ” has been revised to “ rinsing”. (page4 , line161)

The word “in ” has been revised to “with”. (page 4, line 161)

The word “dehydrate ” has been revised to “ dehydration”. (page4 , line167)

The word “Fig 2a ” has been revised to “Figure 2a”. (page5 , line175)

The word “Fig 2b ” has been revised to “ Figure 2b”. (page5 , line176)

The word “ elevation” has been revised to “increase”. (page 5, line184)

The word “ (Fig 4a, 4b). ” has been revised to “ (Figure 4a, 4b).”. (page6 , line199)

The word “ showing” has been revised to “ showed”. (page6 , line214)

The word “displaying ” has been revised to “ displayed”. (page6 , line 215)

The word “ Fig 6a, 6b” has been revised to “Figure 6a, 6b”. (page 7, line 225)

The word “ elevate” has been revised to “inducing”. (page9 , line271)

The word “ elevate” has been revised to “ elevated”. (page9 , line276)

The word “elevate ” has been revised to “ inducing”. (page10 , line292)

The phrase “ the results of Subcutaneous ” has been revised to “Subcutaneous” . (page 10, line291)

The word " indicating " has been revised to " suggesting". (page10 , line294)

Question 5 The authors are encouraged to review the formatting of the cited references to ensure consistency. All references, including journal titles and author names, should adhere uniformly to the journal's prescribed style guidelines.

Reply 5:Thank you for your comments on the relevance of the references in our manuscript. We carefully reviewed each reference to ensure that it directly supported the content and conclusions of our study. Additional new references are listed subsequently, and the format of the literature has been changed to comply with journal requirements.

For example, isocoumarin, a flavonoid derived from carbohydrates, can be synthesised from 3-glycosyl isocoumarin and 3-glucosyl-substituted isocoumarin[16]. Isocoumarins have been demonstrated to possess potent antioxidant and anti-inflammatory activities, which are essential for the treatment of chronic diseases[17].(pages 2, line 57-60)

When the tumors grew to a size of 100 mm3, the mice were randomly divided into two groups: the experimental group (ROF, 10 mg/kg[22], 100 μL, n=5) and control group (1% DMSO, 100 μL, n=5) .(pages 4, line 154)

Reference in the manuscript

[16]Sudarshan, K. and I.S. Aidhen, Convenient Synthesis of 3-Glycosylated Isocoumarins.2017. 2017(1): p. 34-38.

[17]Ramanan, M., et al., Inhibition of the enzymes in the leukotriene and prostaglandin pathways in inflammation by 3-aryl isocoumarins. Eur J Med Chem, 2016.124: p. 428-434.

[22]Chen, H., et al., Rhoifolin ameliorates osteoarthritis via the Nrf2/NF-κB axis: in vitro and in vivo experiments. Osteoarthritis Cartilage, 2022. 30(5): p. 735-745.

Reviewer 2 Report

Comments and Suggestions for Authors

The paper by Chen R. et al. provides a powerful example of the use of traditional Chinese medicine in cancer treatment. As shown by the authors a flavonoid Rhoifolin (ROF)  significantly increased apoptosis in hepatocellular carcinoma (HCC) cell lines HEPG2 and HuH7 by up-regulating production of proteins from Bcl-2 family like BID,BAX , BAK1, Casp8, Casp9 etc and inhibit proliferation of carcinoma cells by arresting them in the S-phase of the cycle. The effects are dose-dependent and IC50 are estimated. It is important that ROF also suppressed growth of HCC in vivo after subcutaneous tumor transplantation into nude mice.

            At the same time, I would like to draw the authors' attention to some points that worsen the quality of the article. There are inconsistencies in Fig 2. What viability is considered as 100%? The authors should explain why the curve of viability of L02 cells starts from 110%. And each point must show the deviation between viability in 3 series of experiments.

            English language is readable but should be corrected. Elevation/ elevate is used too often instead of increase. Example (line 174) “cells demonstrated a notable elevation in apoptotic cells following a 48 hour treatment period”. “The cells showed a marked increase in the number of apoptotic cells after 48 hours of treatment” is better. (line 210) “demonstrated considerable elevation in response to ROF treatment” – “demonstrated a significant increase in ROF treatment responses” is better. (line 53) “with a molecula formula [14] as illustrated in Fig.1 – “a molecular structure of ROF is shown in Fig.1”  etc. A correction of English is necessary. Besides Table 2 line 251 “efect” – should be corrected to effect.

            The paper can be published when the mentioned comments will be taken into account and corrected.

Comments on the Quality of English Language

English language is readable but should be corrected. Elevation/ elevate is used too often instead of increase. Example (line 174) “cells demonstrated a notable elevation in apoptotic cells following a 48 hour treatment period”. “The cells showed a marked increase in the number of apoptotic cells after 48 hours of treatment” is better. (line 210) “demonstrated considerable elevation in response to ROF treatment” – “demonstrated a significant increase in ROF treatment responses” is better. (line 53) “with a molecula formula [14] as illustrated in Fig.1 – “a molecular structure of ROF is shown in Fig.1”  etc. A correction of English is necessary. Besides Table 2 line 251 “efect” – should be corrected to effect.

Author Response

Dear Reviewer,

Attached please find the revised version of our manuscript entitled “Rhoifolin suppresses cell proliferation and induces apoptosis in hepatocellular carcinoma cells in vitro and in vivo” (pharmaceuticals-3379644). We appreciated very much for valuable comments and helpful suggestions from you and the reviewers, which have guided us to significantly improve the quality of our manuscript. We have thoroughly revised the manuscript accordingly, and the changes were highlighted in blue in the revised version.

Question 1. The paper by Chen R. et al. provides a powerful example of the use of traditional Chinese medicine in cancer treatment. As shown by the authors a flavonoid Rhoifolin (ROF) significantly increased apoptosis in hepatocellular carcinoma (HCC) cell lines HEPG2 and HuH7 by up-regulating production of proteins from Bcl-2 family like BID,BAX , BAK1, Casp8, Casp9 etc and inhibit proliferation of carcinoma cells by arresting them in the S-phase of the cycle. The effects are dose-dependent and IC50are estimated. It is important that ROF also suppressed growth of HCC in vivo after subcutaneous tumor transplantation into nude mice.

Reply 1:Thank you very much for your recognition.

Question 2. At the same time, I would like to draw the authors' attention to some points that worsen the quality of the article. There are inconsistencies in Fig 2. What viability is considered as 100%? The authors should explain why the curve of viability of L02 cells starts from 110%. And each point must show the deviation between viability in 3 series of experiments.

Reply 2:Thank you very much for you suggestions.

In the proliferation experiment, the cell numbers of DMSO group was identified as 100%. On this basis, the cell numbers of ROF group was converted into a relative percentage compared with those of DMSO group. We have displayed the experimental point of DMSO group in the revised Figure 2 and supplemented above information in the method section.

We are sorry about that the original standard errors were not very clear, and we have adjusted the format of standard errors to make them more clearly displayed in the revised Figure 2.

Question 3. English language is readable but should be corrected. Elevation/ elevate is used too often instead of increase. Example (line 174) “cells demonstrated a notable elevation in apoptotic cells following a 48 hour treatment period”. “The cells showed a marked increase in the number of apoptotic cells after 48 hours of treatment” is better. (line 210) “demonstrated considerable elevation in response to ROF treatment” – “demonstrated a significant increase in ROF treatment responses” is better. (line 53)

Reply 3:Thanks to your suggestion, we have changed this part of the text. The changes are as follows,

The word “ wash” has been revised to “washed”. (page3 , line109)

The word “detected ” has been revised to “analyzed”. (page3 , line111)

The word “elevate” has been revised to “inducing”. (page9 , line271)

The word “elevate” has been revised to “ elevated”. (page9 , line276)

The word “elevate ” has been revised to “ inducing”. (page10 , line292)

Question 4. “with a molecula formula [14] as illustrated in Fig.1” – “a molecular structure of ROF is shown in Fig.1”  etc. A correction of English is necessary.

Reply 4:Thanks to your suggestion, we have changed this part of the text. The changes are as follows,

The phrase "with a molecula formula [14] as illustrated in Fig.1" has been revised to “is shown in Figure 1”. (page 2, line 61)

The word “Fig 2a ” has been revised to “Figure 2a”. (page5 , line175)

The word “Fig 2b ” has been revised to “ Figure 2b”. (page5 , line176)

The word “ (Fig 4a, 4b). ” has been revised to “ (Figure 4a, 4b).”. (page6 , line199)

The word “ Fig 6a, 6b” has been revised to “Figure 6a, 6b”. (page 7, line 225)

Question 5. Besides Table 2 line 251 “efect” – should be corrected to effect. The paper can be published when the mentioned comments will be taken into account and corrected.

Reply 5:Thanks to your suggestion, we have changed this part of the text. The changes are as follows,

The word “efect” in Table 2 has been revised to “effective”. (page 8, line 252)

The word “promotion ” has been revised to “promoting”. (page2 , line 72)

The word “through ” has been revised to “validated in”. (page2 , line 75)

The word “have ” has been revised to “had”. (page2 , line 85)

The word “using ” has been revised to “with”. (page3 , line 97)

The phrase " wall behind the " has been revised to “were seeded at”. (page3 , line 104)

The phrase " and the " has been revised to “after which”. (page3 , line107)

The word “staining ” has been revised to “stained”. (page3 , line118)

The phrase “was used to detect ” has been revised to “ detection”. (page3 , line121)

The word “rinsed ” has been revised to “ rinsing”. (page4 , line161)

The word “in ” has been revised to “with”. (page 4, line 161)

The word “dehydrate ” has been revised to “ dehydration”. (page4 , line167)

The word “ elevation” has been revised to “increase”. (page 5, line184)

The word “ showing” has been revised to “ showed”. (page6 , line214)

The word “displaying ” has been revised to “ displayed”. (page6 , line 215)

The phrase “ the results of Subcutaneous ” has been revised to “Subcutaneous”. (page 10, line291)

The word " indicating " has been revised to “suggesting”. (page10 , line294)